# Perfectly Secure Steganography Using Minimum Entropy Coupling

**Christian Schroeder de Witt**[*]
FLAIR, University of Oxford
cs@robots.ox.ac.uk

**Samuel Sokota**[*]
Carnegie Mellon University
ssokota@andrew.cmu.edu

**J. Zico Kolter**
Carnegie Mellon University
zkolter@cs.cmu.edu

**Jakob Foerster**
FLAIR, University of Oxford
jakob@robots.ox.ac.uk

**Martin Strohmeier**
armasuisse Science + Technology
martin.strohmeier@armasuisse.ch

## Abstract

Steganography is the practice of encoding secret information into innocuous content in such a manner that an adversarial third party would not realize that there is hidden meaning. While this problem has classically been studied in security literature, recent advances in generative models have led to a shared interest among security and machine learning researchers in developing scalable steganography techniques. In this work, we show that a steganography procedure is perfectly secure under Cachin (1998)'s information theoretic-model of steganography if and only if it is induced by a coupling. Furthermore, we show that, among perfectly secure procedures, a procedure is maximally efficient if and only if it is induced by a minimum entropy coupling. These insights yield what are, to the best of our knowledge, the first steganography algorithms to achieve perfect security guarantees with non-trivial efficiency; additionally, these algorithms are highly scalable. To provide empirical validation, we compare a minimum entropy coupling-based approach to three modern baselines—arithmetic coding, Meteor, and adaptive dynamic grouping—using GPT-2, WaveRNN, and Image Transformer as communication channels. We find that the minimum entropy coupling-based approach achieves superior encoding efficiency, despite its stronger security constraints. In aggregate, these results suggest that it may be natural to view information-theoretic steganography through the lens of minimum entropy coupling.

## 1 Introduction

In steganography (Blum & Hopper, 2004; Cachin, 2004), the goal, informally speaking, is to encode a *plaintext* message into another form of content (called *stegotext*) such that it appears similar enough to innocuous content (called *covertext*) that an adversary would not realize that there is hidden meaning. Because steganographic procedures hide the existence of sensitive communication altogether, they provide a complementary kind of security to that of cryptographic methods, which only hide the contents of the sensitive communication—not the fact that it is occurring.

In this work, we consider the information-theoretic model of steganography introduced in (Cachin, 1998). In Cachin (1998)'s model, the exact distribution of covertext is assumed to be known to all parties. Security is defined in terms of the KL divergence between the distribution of covertext and the distribution of stegotext. A procedure is said to be perfectly secure if it guarantees a divergence of zero. Perfect security is a very strong notion of security, as it renders detection by statistical or

---

[*]Equal contribution

human analysis impossible. To the best of our knowledge, the only existing algorithms that achieve both perfect security and non-trivial efficiency are limited to specific distributions of covertext, such as the uniform distribution, making their applicability limited.

The main contribution of this work is formalizing a relationship between perfect security and couplings of distributions—that is, joint distributions that marginalize to prespecified marginal distributions. We provide two results characterizing this relationship. First, we show that a steganographic procedure is perfectly secure if and only if it is induced by couplings between the distribution of *ciphertext* (an encoded form of plaintext that can be made to look uniformly random) and the distribution of covertext. Second, we show that, among perfectly secure procedures, a procedure is maximally efficient if and only if it is induced by couplings whose joint entropy are minimal—that is, minimal entropy couplings (MECs) (Kovačević et al., 2015).

While minimum entropy coupling is an NP-hard problem, there exist $O(N \log N)$ approximation algorithms (Kocaoglu et al., 2017; Cicalese et al., 2019; Rossi, 2019) that are suboptimal (in terms of joint entropy) by no more than one bit, while retaining exact marginalization guarantees. Furthermore, Sokota et al. (2022) recently introduced an iterative minimum entropy coupling approach, which we call iMEC, that iteratively applies these approximation procedures to construct couplings between one uniform distribution and one autoregressively specified distribution, both having arbitrarily large supports, while still retaining marginalization guarantees. We show that, because ciphertext can be made to look uniformly random, and any distribution of covertext can be specified autoregressively, iMEC can be leveraged to perform steganography with arbitrary covertext distributions and plaintext messages. *Excitingly, to the best of our knowledge, this represents the first instance of a steganography algorithm with non-trivial efficiency and perfect security guarantees that scales to arbitrary distributions of covertext.*

In our experiments, we evaluate iMEC using GPT-2 (Radford et al., 2019) (a language model), WaveRNN (Kalchbrenner et al., 2018) (an audio model), and Image Transfomer (an image model) as covertext distributions. We compare against arithmetic coding (Ziegler et al., 2019), Meteor (Kaptchuk et al., 2021), and adaptive dynamic grouping (ADG) (Zhang et al., 2021), other recent methods for performing information theoretic-steganography with deep generative models. To examine empirical security, we estimate the KL divergence between the stegotext and the covertext for each method. For iMEC, we find that the KL divergence is on the order of numerical precision, in agreement with our theoretical guarantees. In contrast, arithmetic coding, Meteor, and ADG yield KL divergences many orders of magnitude larger, reflecting their weaker security guarantees. To examine encoding efficiency, we measure the number of bits transmitted per step. We find that iMEC generally yields superior efficiency results to those of arithmetic coding, Meteor, and ADG, despite its stricter constraints.

We would summarize our theoretical results as showing that minimum entropy coupling-based approaches are the most efficient perfect security approaches and our empirical results as showing that minimum entropy coupling-based approaches can be more efficient than less secure alternatives. In aggregate, we believe that these findings suggest that it may be natural to view information-theoretic steganography through the perspective of minimum entropy coupling.

## 2 BACKGROUND

In the first half of this section, we review the information-theoretic model of steganography introduced by Cachin (1998). In the second half, we review couplings and minimum entropy couplings (Kovačević et al., 2015).

### 2.1 AN INFORMATION-THEORETIC MODEL FOR STEGANOGRAPHY

**Problem Setting** The objects involved in information-theoretic steganography can be divided into two classes: those which are externally specified and those which require algorithmic specification. Each class contains three objects. The externally specified objects include the distribution over plaintext messages $\mathcal{M}$, the distribution over covertext $\mathcal{C}$, and the random source generator.

- The distribution over plaintext messages $\mathcal{M}$ may be known by the adversary, but is not known by the sender or the receiver. However, the sender and receiver are aware of the domain $\mathbb{M}$ over which

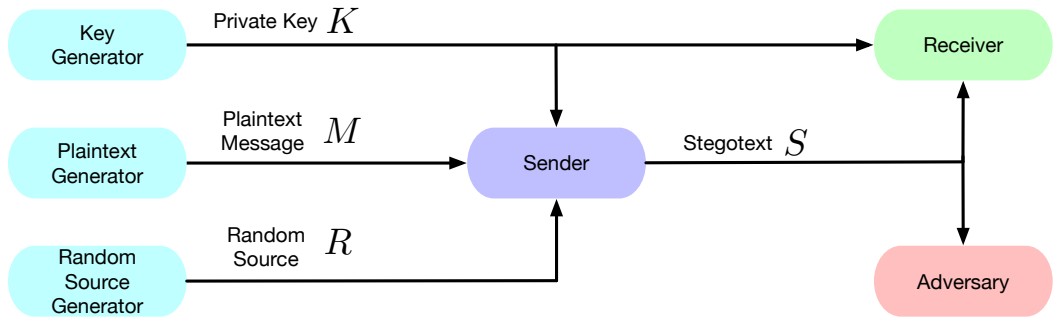

Figure 1: A graphical depiction of steganography. The sender receives a plaintext message, a source of randomness, and a private key, and outputs a stegotext. The receiver receives the same private key as the sender, along with the stegotext. The adversary also receives the stegotext.

$\mathcal{M}$ ranges. The sampled plaintext message $M$ is explicitly known by the sender, but not to the receiver or the adversary.

- The covertext distribution $\mathcal{C}$ is assumed to be known by the sender, the receiver, and the adversary.

- The random source generator provides the sender with a mechanism to take random samples from distributions. This random source is known to the sender but not to the receiver or adversary.

The objects requiring algorithmic specification, which are collectively referred to as a stegosystem, are the key generator, the encoder, and the decoder.

- The key generator produces a private key $K$ in the form of a binary string. This private key is shared between the sender and receiver over a secure channel prior to the start of the stegoprocess and can be used to coordinate encryption and decryption. The key generation process may be known to the adversary, but the realization of the key $K$ is not.

- The encoder takes a private key $K$, a plaintext message $M$, and a source of randomness $R$ as input and produces a stegotext $S$ in the space of covertexts $\mathbb{C}$.

- The decoder takes a private key $K$ and a stegotext $S$ as input and returns an estimated plaintext message $\hat{M}$.

Many of the objects described above are depicted visually in Figure 1.

In this work, we are interested in stegosystems that possess perfect security.

**Definition 2.1.** *(Cachin, 1998) Given covertext distribution $\mathcal{C}$ and plaintext message space $\mathbb{M}$, a stegosystem is $\epsilon$-secure against passive adversaries if the KL divergence between the distribution of covertext $\mathcal{C}$ and the distribution of stegotext $\mathcal{S}$ less than $\epsilon$; i.e., $KL(\mathcal{C}, \mathcal{S}) < \epsilon$. It is perfectly secure if the KL divergence is zero; i.e., $KL(\mathcal{C}, \mathcal{S}) = 0$.*

In other words, a steganographic system is perfectly secure if the distribution of stegotext $\mathcal{S}$ communicated by the sender is exactly the same as the distribution of covertext $\mathcal{C}$.

In addition to security, it is desirable for stegosystems to transmit information efficiently. Mutual information between messages and stegotexts is one way to quantify efficiency.

**Definition 2.2.** *The mutual information $\mathcal{I}(M; S) = \mathcal{H}(M) - \mathcal{H}(M \mid S)$ between the message $M$ and stegotext $S$ is the expected amount of uncertainty in the message $M$ that is removed by conditioning on the stegotext $S$.*

**Methodological Outline** A common class of stegosystems uses two-step encoding and two-step decoding processes, as described below:

1. The sender uses the private key $K$ to injectively map the plaintext message $M$ into ciphertext $\mathbb{X} = \{0,1\}^\ell$ in such a way that the induced distribution over ciphertext $\mathcal{X}$ is uniformly random, regardless of the distribution of $\mathcal{M}$.[1]

2. The sender uses a (potentially stochastic) mapping $f: \mathbb{X} \rightsquigarrow \mathbb{C}$ to transform the ciphertext $X$ into stegotext $S$ (which exists in the space of covertexts $\mathbb{C}$).

3. The receiver estimates the ciphertext $\hat{X}$ from the stegotext $S$.

4. The receiver inverts the estimated ciphertext $\hat{X}$ to a plaintext message $\hat{M}$ with private key $K$.[2]

In the outline above, steps 1 and 4 can be accomplished using standard operations in steganography literature and, thus, are left implicit in much of the remainder of the paper. Our methodological contribution specifically concerns steps 2 and 3.

## 2.2 COUPLING

The central tool we will use to approach steganography is based on coupling distributions. Let $\mathcal{X}$ and $\mathcal{Y}$ be probability distributions over finite sets $\mathbb{X}$ and $\mathbb{Y}$. A coupling $\gamma$ of $\mathcal{X}$ and $\mathcal{Y}$ is a joint distribution over $\mathbb{X} \times \mathbb{Y}$ such that, for all $x \in \mathbb{X}, \sum_{y'} \gamma(x, y') = \mathcal{X}(x)$ and such that, for all $y \in \mathbb{Y}, \sum_{x'} \gamma(x', y) = \mathcal{Y}(y)$. In other words, a coupling is a joint distribution over $\mathbb{X}$ and $\mathbb{Y}$ that marginalizes to $\mathcal{X}$ and $\mathcal{Y}$, respectively. Note that, in general, there may be many possible couplings for distributions $\mathcal{X}$ and $\mathcal{Y}$.

**Minimum Entropy Coupling** Let $\Gamma(\mathcal{X}, \mathcal{Y})$ denote the set of all couplings. A minimum entropy coupling (MEC) is an element of $\Gamma(\mathcal{X}, \mathcal{Y})$ with minimal entropy. In other words, a MEC is $\gamma \in \Gamma(\mathcal{X}, \mathcal{Y})$ such that the joint entropy $\mathcal{H}(\gamma) = -\sum_{x,y} \gamma(x, y) \log \gamma(x, y)$ is no larger than that of any other coupling $\gamma' \in \Gamma(\mathcal{X}, \mathcal{Y})$.

## 3 STEGANOGRAPHY AND MINIMUM ENTROPY COUPLING

We are now ready to explain our main results. First, we define what it means for an encoding procedure to be induced by a coupling.

**Definition 3.1.** *We say that an encoding procedure $f: \mathbb{X} \rightsquigarrow \mathbb{C}$ is induced by a coupling if there exists $\gamma \in \Gamma(\mathcal{X}, \mathcal{C})$ such that for all $x \in \mathbb{X}, c \in \mathbb{C}, \mathcal{P}(f(x){=}c) = \gamma(C{=}c \mid X{=}x)$.*

We formalize a relationship between steganographic encoding procedures with perfect security and steganographic encoding procedures that are induced by couplings.

**Theorem 1.** *A steganographic encoding procedure is perfectly secure if and only if it is induced by a coupling.*

*Proof.* ($\Rightarrow$) Assume that steganographic encoding procedure $f: \mathbb{X} \rightsquigarrow \mathbb{C}$ is perfectly secure. Define a marginal probability distribution $\gamma(X{=}x) = \mathcal{X}(x)$ and a conditional probability distribution $\gamma(C{=}c \mid X{=}x) = \mathcal{P}(f(x){=}c)$. Note that, together, $\gamma(X)$ and $\gamma(C \mid X)$ define a joint probability distribution $\gamma(X{=}x, C{=}c) = \gamma(X{=}x)\gamma(C{=}c \mid X{=}x)$. It suffices to show that this joint distribution is a coupling. To see this, first, consider that, by construction, marginalizing over $C$ yields $\mathcal{X}$. Second, note that, by the definition of perfect security, for all $c \in \mathbb{C}$, we must have $\mathbb{E}_{X \sim \mathcal{X}} \mathcal{P}(f(X) = c) = \mathcal{C}(c)$. Thus, because $\mathcal{P}(f(X){=}c) = \gamma(C{=}c \mid X)$, marginalizing over $\mathbb{X}$ must yield $\mathcal{C}$.

($\Leftarrow$) Assume that there is some $\gamma \in \Gamma(\mathcal{X}, \mathcal{C})$ that induces steganographic encoding procedure $f: \mathbb{X} \rightsquigarrow \mathbb{C}$. Then, by definition, for all $c \in \mathbb{C}$, we must have $\mathbb{E}_{X \sim \mathcal{X}} \gamma(C{=}c \mid X) = \mathcal{C}(c)$. Thus, because $\mathcal{P}(f(X){=}c) = \gamma(C{=}c \mid X)$, it follows that $f$ is perfectly secure. □

Next, we formalize a relationship between maximally efficient perfectly secure steganographic encoding procedures and stegagraphic encoding procedures induced by minimum entropy couplings.

---

[1] For example, if $K$ is drawn from a uniform random distribution, $\text{bin}(M)$ denotes a deterministic binarization of $M$, and XOR represents the component-wise exclusive-or function, then $X = \text{XOR}(\text{bin}(M), K)$ is guaranteed to be distributed uniformly randomly, regardless of the distribution of messages $\mathcal{M}$.

[2] For the example in footnote 1, the receiver can recover the binarized message $\text{bin}(M)$ using the mapping $X \mapsto \text{XOR}(X, K)$ and invert the binarization to recover the plaintext $M$.

**Theorem 2.** *Among perfectly secure encoding procedures, a procedure $f : \mathbb{X} \rightsquigarrow \mathbb{C}$ maximizes the mutual information $\mathcal{I}(M; S)$ if and only if $f$ is induced by a minimum entropy coupling.*

*Proof.* First, note that, because the mapping from plaintext messages is injective, we have $\mathcal{I}(M; S) = \mathcal{I}(X; S)$. Thus, it suffices to prove the analogous statement for $\mathcal{I}(X; S)$. Also, note that, because the theorem's statement is specific to perfectly secure $f$, we know from Theorem 1 that we need only concern ourselves with procedures induced by $\gamma \in \Gamma(\mathcal{X}, \mathcal{C})$.

($\Rightarrow$) Assume that $\gamma$ maximizes $\mathcal{I}(X; S)$. Then, because $\mathcal{I}(X; S) = \mathcal{H}(X) + \mathcal{H}(S) - \mathcal{H}(X, S)$, $\gamma$ must also maximize the right-hand side of the equality. Furthermore, because $\mathcal{H}(X)$ is fixed and $\mathcal{H}(S)$ is equal to $\mathcal{H}(C)$ by definition of perfect security, it follows that $\gamma$ must minimize $\mathcal{H}(X, S)$. Then, by definition, $\gamma$ is a minimum entropy coupling.

($\Leftarrow$) Assume that $\gamma$ is a minimum entropy coupling. Then, because $\mathcal{H}(X; S) = \mathcal{H}(X) + \mathcal{H}(S) - \mathcal{I}(X, S)$, $\gamma$ must also minimize the right-hand side of the equality. Furthermore, because $\mathcal{H}(X)$ is fixed and $\mathcal{H}(S)$ is equal to $\mathcal{H}(C)$ by definition of perfect security, it follows that $\gamma$ must maximize $\mathcal{I}(X, S)$. □

Informally, Theorem 1 can be viewed as stating that the class of perfectly secure stenographic procedures and the class of set of couplings are equivalent; Theorem 2 can be viewed as stating that the subclass of perfectly secure procedures that maximize mutual information and the set of minimum entropy couplings are equivalent. On one hand, these theorems illuminate an avenue toward achieving efficient and perfectly secure steganography. On the other hand, they show that any steganographic procedures that achieve perfect security and that any steganographic procedures that achieve perfect security with maximal efficiency do so only by virtue of the fact that they are induced by couplings and minimum entropy couplings, respectively.

## 4 A Minimum Entropy Coupling-Based Approach to Steganography

To turn the insights from the previous section into an actionable steganography algorithm, we require MEC techniques. While computing an exact MEC is an NP-hard problem, Cicalese et al. (2019) and Rossi (2019) recently showed that it is possible to approximate MECs in $N \log N$ time with a solution guaranteed to be suboptimal by no more than one bit. These algorithms are fast enough to compute low entropy couplings for distributions with small supports. Even more recently, Sokota et al. (2022) introduced an iterative minimum entropy coupling (iMEC) approach that uses an approximate MEC algorithm as a subroutine to iteratively couple distributions with arbitrarily large supports, so long as one of the distributions is uniform and the other can be specified autoregressively. This approach provably produces couplings, meaning that exact marginalization (and therefore perfect security) is guaranteed. We describe how iMEC can be directly applied as a steganography algorithm below.

**Iterative Minimum Entropy Coupling** Assume that $\mathcal{X}$ is a uniform distribution and let $\mathbb{X}_1 \times \cdots \times \mathbb{X}_n = \mathbb{X}$ and $\mathbb{C}_1 \times \cdots \times \mathbb{C}_m = \mathbb{C}$ be factorizations over the spaces that $\mathcal{X}$ and $\mathcal{C}$ range. iMEC implicitly defines a coupling $\gamma$ between $\mathcal{X}$ and $\mathcal{C}$ using procedures that iteratively call an approximate MEC algorithm as a subroutine. These procedures can sample $S \sim \gamma(C \mid X{=}x)$ and compute $\gamma(X \mid C{=}s)$ for a given $x$ and $s$, respectively. We will call these operations encoding and decoding.

---

**Algorithm 1** Sampling $S \sim \gamma(C \mid X{=}x)$

**procedure** ENCODE$(x, \mathcal{X}, \mathcal{C})$
    Initialize a uniform distribution $\mu_i$ over $\mathbb{X}_i$ for each $i = 1, \ldots, n$.
    **for** $j = 1, \ldots, m$ **do**
        $i^* \leftarrow \text{argmax}_i \mathcal{H}(\mu_i)$                                      ▷ Select maximal entropy block
        $\gamma_j \leftarrow \text{MEC}(\mu_{i^*}, \mathcal{C}(C_j \mid C_{1:j-1}{=}S_{1:j-1}))$      ▷ Call approximate MEC subroutine
        $S_j \sim \gamma_j(C_j \mid X_{i^*}{=}x_{i^*})$                          ▷ Sample encoded stegotoken
        $\mu_{i^*} \leftarrow \gamma_j(X_{i^*} \mid C_j{=}S_j)$                       ▷ Update ciphertext posterior
    **end for**
    return $S$                                                  ▷ Return stegotext
**end procedure**

---

---

**Algorithm 2** Computing $\gamma(X \mid C{=}s)$

---

**procedure** DECODE$(s, \mathcal{X}, \mathcal{C})$
    Initialize a uniform distribution $\mu_i$ over $\mathbb{X}_i$ for each $i = 1, \ldots, n$.
    **for** $j = 1, \ldots, m$ **do**
        $i^* \leftarrow \arg\max_i \mathcal{H}(\mu_i)$                         ▷ Select maximal entropy block
        $\gamma_j \leftarrow \text{MEC}(\mu_{i^*}, \mathcal{C}(C_j \mid C_{1:j-1}{=}s_{1:j-1}))$     ▷ Call approximate MEC subroutine
        $\mu_{i^*} \leftarrow \gamma_j(X_{i^*} \mid C_j{=}s_j)$                ▷ Update ciphertext posterior
    **end for**
    return $x \mapsto \prod_i \mu_i(x_i)$                      ▷ Return ciphertext posterior
**end procedure**

---

Encoding, described in Algorithm 1, and decoding, described in Algorithm 2, share similar structures. Encoding greedily encodes information about $x$ into $S$, using the induced posterior over $\mathbb{X}$ to maximize coupling efficiency; decoding simply reproduces this posterior and selects the maximum a posteriori.

## 5 EXPERIMENTS

To validate iMEC as a steganography algorithm, we evaluate it empirically against arithmetic coding (Ziegler et al., 2019), Meteor (Kaptchuk et al., 2021), and ADG (Zhang et al., 2021) on four different covertext types. We also include a variant of Meteor that employs bin-sorted probabilities that we call Meter:reorder.

### 5.1 EXPERIMENT SETUP

Our first covertext distribution consists of uniformly random noise of dimension $40$ and a mean covertext entropy of $\overline{\mathcal{H}}_C = 5.32$ bits per token. Our other channels are designed to mimic the natural distributions of three widely utilized modes of human communication: text, speech, and images. The second and third covertext distributions are variants of GPT-2 (Radford et al., 2019) with 12 attention modules (Wolf et al., 2020) conditioned on $1024$-character strings from the *Wikitext-103* dataset (Merity et al., 2016). The second performs *top-k* sampling from a re-normalized categorical distribution over the $40$ highest-probability outputs. The third performs *nucleus sampling* (Holtzman et al., 2020) from the highest-probability outputs that together comprise $90\%$ of the raw channel entropy (Radford et al., 2019). The fourth covertext distribution consists of a *text-to-speech (TTS) pipeline* (Yang et al., 2022) based on Tacotron-2 (Shen et al., 2018) mel-spectogram encodings, followed by a WaveRNN generative model (Kalchbrenner et al., 2018) pretrained on the LJSpeech dataset (Ito & Johnson, 2017) with a mean covertext entropy of $\overline{\mathcal{H}}_C = 0.35$ bits per token. We condition the WaveRNN model using text generated by GPT-2 that we convert into audio data using mel-spectogram encodings. The fifth covertext distribution is over an unconditional transformer-based generative model (Parmar et al., 2018) of *CIFAR-10* (Krizhevsky et al., 2009) RGB images of dimensions $32 \times 32$.

Our implementation of iMEC makes use of the approximative minimum entropy coupling procedure described in Algorithm 1 of (Kocaoglu et al., 2017) as its subprocedure. We show results for three different iMEC block sizes (10 bits, 16 bits, and 20 bits). The number of bits measures the initial entropy of a particular $\mathcal{H}(\mu_i)$. The larger the block size, the more burden iMEC places on the approximate MEC subroutine—this may result in lower entropy couplings but have a higher computational cost. We used the original implementations of arithmetic coding (Ziegler et al., 2019), Meteor/Meteor:reorder (Kaptchuk et al., 2021), and ADG (Zhang et al., 2021) for the baselines.

All experiments were performed on a AMD Ryzen Threadripper PRO 3955WX with 16 physical cores and 2x NVIDIA GeForce RTX 3090 GPUs. Apart from model queries, iMEC encoding and decoding occupies just a single CPU core, while arithmetic coding, ADG, and Meteor make use of multiple CPUs, and Meteor:reorder also makes use of GPUs during encoding and decoding.

All ciphertexts are 80-bit bitstrings sampled uniformly at random. We measure the encoding efficiency of each method by measuring the amount of covertext required to transmit these bitstrings. We tuned the hyper-parameters of each method to yield the best performance on this task. For iMEC, we stop

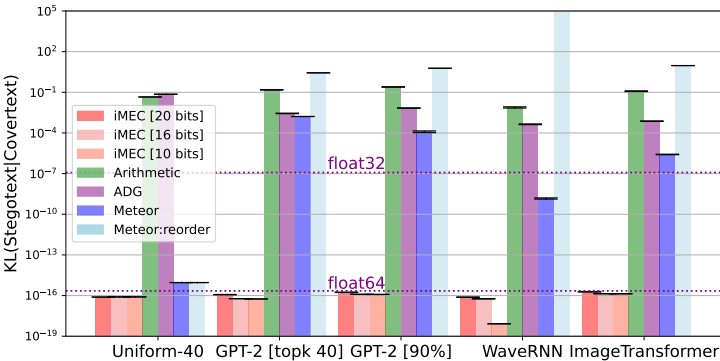

Figure 2: Kullback-Leibler divergences between the stegotext distribution and covertext distribution for each method.

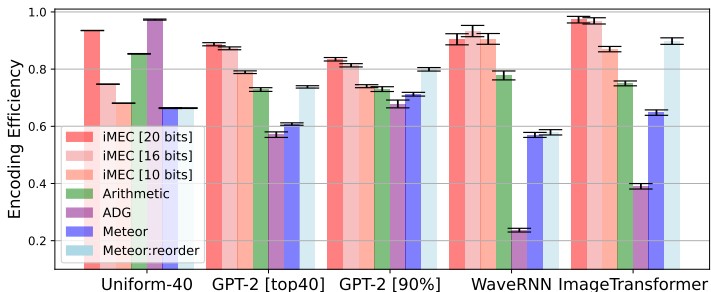

Figure 3: Comparative encoding efficiencies of iMEC with baselines. Each method is evaluated at tuned hyperparameters.

transmission when the entropy of every posterior block is less than $0.1$, i.e., $\forall i, \mathcal{H}(\mu_i) < 0.1$. We found the error rate for this threshold to be lower than $1$ in $10^6$ bits (see Figure 5). For arithmetic coding and Meteor/Meteor:reorder, we performed a grid search over precision values of $8, 16, 32$ and $48$ bits. We find that both arithmetic coding and Meteor get stuck in a seemingly infinite loop about $3\%$ of the time (see Figure 6). For simplicity, we simply discard these runs from our investigation; this may result in overestimation of these algorithms' encoding efficiencies. Errorbars indicate estimates of $95\%$ confidence intervals computed from the central limit theorem over 1000 runs.

## 5.2 RESULTS

**Covertext Divergence** We first investigate the empirical security of each method. We follow the precedent of previous work (Ziegler et al., 2019; Zhang et al., 2021) and show $\text{KL}(\mathcal{S}, \mathcal{C})$, which can easily be estimated from samples. We present the results in Figure 2. We used float64s for the experiments; thus, we expect the divergence of methods with perfect security should generally be on a similar order of magnitude as float64s. Overall, our results suggest that that is true—iMEC's KL divergence is upper bounded by numerical precision in every setting. On the other hand, arithmetic coding and ADG possess divergences that are consistently many orders of magnitude above the precision of the data type. While Meteor and Meteor:reorder exhibit relatively better performance on uniform covertext distributions, they incur larger divergences for the complex distributions (including a divergent KL divergence for Meteor:reorder on WaveRNN).

**Encoding Efficiency** We next compare encoding efficiencies and bit rates. We say bit rate to mean the number of bits encoded per token. We say encoding efficiency to mean the bit rate normalized to the mean covertext entropy per token. We show the results for encoding efficiencies in Figure 3; the

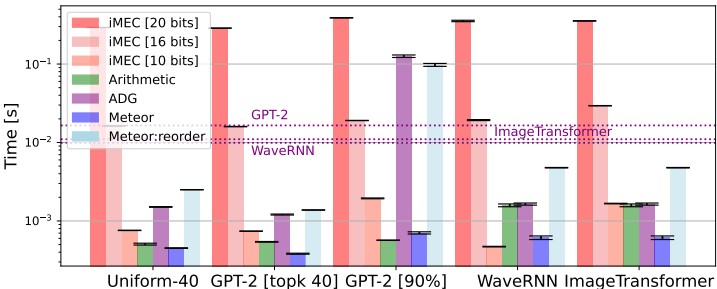

Figure 4: Comparative speed evaluation of iMEC and baselines. Horizontal lines indicate the amount of time required for model inference for GPT-2, WaveRNN, and Image Transformer.

results for bit rate are shown in Figure 7 in the appendix. Surprisingly, despite its perfect security constraints, iMEC with a block size of 20 bits yields superior encoding efficiency to the baselines across all three of our realistic distributions. We also observe that iMEC with a block size of 16 bits generally yields competitive or superior performance to the other methods, and that 10 bit block sizes also performs competitively. Impressively, at a standard encoding frequency of 24kHz, iMEC's performance for WaveRNN would allow it to encode a private message of 225 kilobytes in just 30 seconds of TTS voicemail—sufficient for sending compressed images.

Performance among the baselines varies. Arithmetic coding achieves greater efficiency than iMEC[10 bits] and iMEC[16 bits] on the uniform covertext distribution and comparable performance with iMEC[10 bits] on GPT-2 90%. ADG achieves the best efficiency of all methods on the uniform covertext distribution, but is the least efficient otherwise, especially on WaveRNN. Meteor is generally less efficient than all iMEC variants and arithmetic coding. Meteor:reorder achieves greater efficiency than iMEC[10 bits] for GPT-2 90%, but is otherwise less efficient than all iMEC variants.

**Speed** Lastly, we examine the speed of each algorithm. We show results in Figure 4. While, in the previous section, we observed that increasing iMEC's block size can improve encoding efficiency, we see here that this improved efficiency does not come without cost. While 10 bit blocks require an order of magnitude less time than model inference, 16 bit blocks require the same order of magnitude of time as model inference, and 20 bit blocks require an order of magnitude more time than that. The wall-clock time of arithmetic coding and Meteor are generally comparable to that of the 10 bit blocks, while the wall-clock of Meteor:reorder and ADG vary somewhat significantly depending on the task. We believe it is possible (perhaps even likely) that innovations in approximate minimum entropy coupling will allow some of the cost of coupling to be distributed across multiple cores, making the block sizes we experiment with here much cheaper.

## 6 RELATED WORK

**Provable Perfect Security** One body of related work concerns algorithms with provably perfect security. One case in which perfect security is possible is when the covertext distribution is uniform (Cachin, 1998). In this case, perfect security can be achieved by embedding the message in uniform random ciphertext over the same domain as the covertext. However, constructing algorithms that both guarantee perfect security and transmit information at non-vanishing rates for more general covertext distributions has proved challenging. One notable result is that of Wang & Moulin (2008), who show that, in the case that tokens of the covertext are independently identically distributed, public watermarking codes (Moulin & O'Sullivan, 2003; Somekh-Baruch & Merhav, 2003; 2004) that preserve first order statistics can be used to construct perfectly secure steganography protocols of the same error rate. Another relevant line of research is that of Ryabko & Ryabko (2009), who show that perfectly secure steganography can be achieved in the case that letters of the covertext are independently identically distributed under the weaker assumption of black box access to the covertext distribution. In follow up work, Ryabko & Ryabko (2011) generalize their earlier work to a setting in which the letters of the covertext need only follow a $k$-order Markov distribution. Unlike

these works, our approach does not make any assumptions on the structure of the distribution of covertext, though, unlike Ryabko & Ryabko (2011), we do assume that this distribution is known.

**Information-Theoretic Steganography with Deep Generative Models** Another body of related work concerns scaling information-theoretic steganography to covertext distributions represented by deep generative models. Most work in this literature leverages the relationship between compression and steganography (Sallee, 2003), wherein the steganography problem is viewed as a compression problem for the covertext distribution. Existing algorithms in this class are based on Huffman coding (Dai & Cai, 2019) and arithmetic coding (Ziegler et al., 2019; Kaptchuk et al., 2021; Chen et al., 2022). While some of these works possess $\epsilon$-security guarantees, they are not perfectly secure unless the covertext distribution is compressed exactly into the ciphertext distribution, which is not generally the case. Outside of coding-based methods, Zhang et al. (2021) describe another method that they call ADG, which, similarly to coding based approaches, possesses nearly perfect—but not perfect—security guarantees. Our experiments confirm the lack of perfect security guarantees for these methods—indeed, we observed that arithmetic coding (Ziegler et al., 2019), Meteor (Kaptchuk et al., 2021), and ADG (Zhang et al., 2021) incurred KL divergences many orders of magnitude greater than the level of numerical precision we used.

## 7 CONCLUSION AND FUTURE WORK

In this work, we showed 1) that perfect steganographic security is equivalent to a coupling problem and 2) that achieving maximal transmission efficiency among perfect security procedures is equivalent to a minimum entropy coupling problem. These findings may suggest that information-theoretic steganography is viewed most naturally through the lens of minimum entropy coupling. Furthermore, using recent innovations in approximate and iterative minimum entropy coupling (Kocaoglu et al., 2017; Sokota et al., 2022), we showed that this insight is also practical, leading to a MEC-based approach that is scalable to covertext distributions expressed by deep generative models. In empirical evaluations, we found that our MEC-based approach's divergence from the covertext distribution is on the order of numerical precision, in agreement with our theoretical results. Moreover, we observed that it can achieve superior encoding efficiency compared to arithmetic coding-based methods. The existence of a scalable stegosystem with perfect security guarantees and high empirical efficiency may suggest that statistical steganalysis carries little value within Cachin (1998)'s model.

The most significant limitations of our work arise from the assumptions of Cachin (1998)'s information-theoretic formulation of steganography. First, Cachin (1998) assumes that the adversary is passive and that the stegotext arrives to the receiver unperturbed. While this assumption is standard among related work and may hold for many digital transmission channels, it is unrealistic in other settings. One direction for future work is to extend a variant of the minimum entropy coupling perspective of steganography to settings in which the channel medium is noisy. This may be possible by taking inspiration from ideas in error correction literature. Second, Cachin (1998) assumes that white box access to the covertext distribution is available. Unfortunately, this assumption is often unrealistic—and even modern deep generative models struggle to accurately approximate complex distributions (though it is expected that this issue will be somewhat ameliorate over time). Furthermore, in realistic scenarios, the distribution of "normally" occurring content may shift over time and depend on other external context, making it difficult to capture (though it would also affect the adversary's ability to detect distributional anomalies). While there exists information-theoretic steganography literature that does not assume white box access to the covertext distribution under the name universal steganography, it is difficult to achieve general purpose guarantees. Thus, dropping this second assumption may be more challenging, suggesting that a minimum entropy coupling perspective on steganography may be best suited to settings in which the covertext distribution can be modeled accurately.

## 8 REPRODUCIBILITY

An open source implementation of iMEC is available at `https://github.com/schroederdewitt/perfectly-secure-steganography`.

## 9 ACKNOWLEDGMENTS

We thank Philip Torr, Yiding Jiang, Pratyush Maini, and Jeremy Cohen for helpful feedback. This research was supported by the Cooperative AI Foundation, Armasuisse Science+Technology, and the Bosch Center for Artificial Intelligence. CSDW has additionally been supported by an EPSRC IAA Doctoral Impact Fund award kindly hosted by Prof. Philip Torr.

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

# A APPENDIX

## A.1 IMEC ERROR RATE

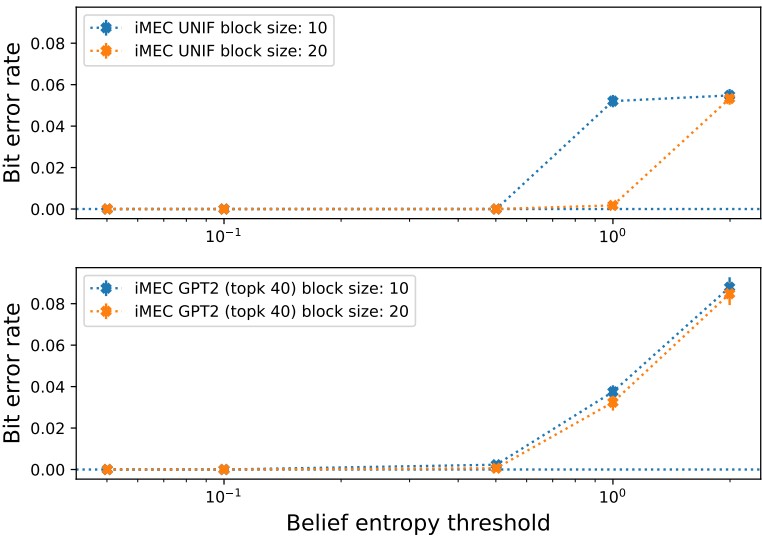

Figure 5: Bit error rate as a function of threshold size. The error bars shown are the standard deviation of the mean over 100 trajectories.

We show error rate as a function of the belief entropy threshold in Figure 5. As is suggested by the figure, the error rate can be made arbitrarily small by selecting a sufficiently small treshold value.

## A.2 NON-TERMINATION FREQUENCY FOR ARITHMETIC CODING

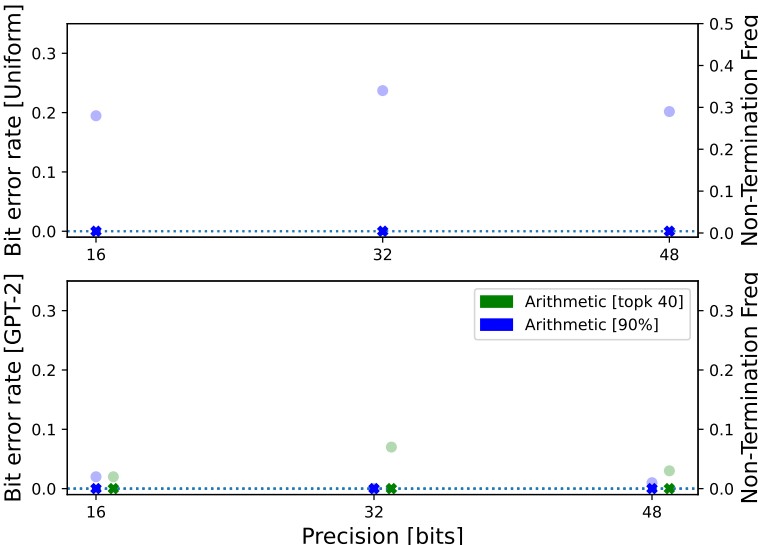

Figure 6: Bit error rate (X) and non-termination frequencies (circles) for arithmetic coding vs. precision hyperparameter. We show estimates over 100 trajectories.

A plot of the error rate and non-termination frequency is shown in Figure 6. While we did not observe errors, non-termination occurred with non-negligible probability.

### A.3    ENCODING BIT RATE

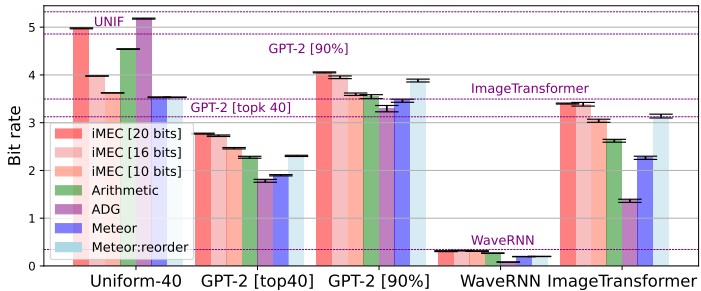

Figure 7: The encoding bit rate for each method for each covertext distribution. Horizontal lines on the right plot correspond to the mean covertext entropy per token.

An unnormalized version of Figure 3 is shown in Figure 7.

### A.4    STEGOTEXT SAMPLES

To illustrate the effect of bias, we reproduce a sample stegotext for both iMEC (block size 10), as well as Meteor:reorder (precision 32) and the private message length is 20 bytes. Both examples have been mildly post-processed for readability, such as removing special characters and whitespaces.

Context:

> Heck horses are dun or grullo (a dun variant) in color, with no white markings. The breed has primitive markings, including a dorsal stripe and horizontal striping on the legs. Heck horses generally stand between 12.2 and 13.2 hands ( 50 and 54 inches, 127 and 137 cm) tall. The head is large, the withers low, and the legs and hindquarters

iMEC produces the following stegotext:

> are short. The neck is wide and thick, a characteristic that can be inherited from the male. The face can be seen as a broad head, with pointed toes. The head and neck are often used as a tool for hunting, though their appearance often depends on their social organization. The legs are

Meteor:reorder produces the following stegotext:

> have a narrow and angular shape. The fore and hind legs are longer than the head. The tail is broad and short in a shape similar to the neck or neckbone. The front legs have a sharp protrusion that leads from the head to the head but not from the tail. The hind legs have long pangs (2) and lower

Note that Meteor:reorder's high bias seemingly lowers the content quality of the output text.

