# OpenReview forum: "Perfectly Secure Steganography Using Minimum Entropy Coupling"
_ICLR.cc/2023/Conference — ICLR 2023 poster_

### Official Review · Reviewer_3EY8 · 2022-10-17

**Confidence:** 3
**Correctness:** 3
**Technical Novelty And Significance:** 3
**Empirical Novelty And Significance:** 3
**Recommendation:** 6

**Clarity, Quality, Novelty And Reproducibility:**

## Clarity
This work is technically sound to the best of my judgment.

## Quality
This work is clearly written and easy to follow.

## Novelty
This work holds novelty for a subgroup of the ML community.

## Reproducibility
This work is reproducible to the best of my judgment.


**Strength And Weaknesses:**

## Strengths
(+) To the best of my knowledge, this is the first work approaching the task of steganography as a minimum entropy coupling problem.
(+) To the best of my knowledge, this is the first steganography algorithm to achieve perfect security guarantees (in the context of Cachin (1998)’s information theoretic model of steganography)
(+) The presented results based on the Kullback-Leibler divergences are convincing (close to zero)
(+) The approach is well motivated by the existing theory.

## Weaknesses
(-) This work relies on the assumption that “the exact distribution of covertext is assumed to be known”. However, in practice, this might not be the case, since the model of language might not be perfectly known. Hence, for most practical scenarios this approach might not provide perfectly secure steganography.
(-) While the results in Figure 2 are close to zero, they are not exactly zero. Does this mean that perfect security is not obtained as claimed?
(-) The authors evaluated the security of their approach empirically, however, it is not clear if this method can withstand modern steganalysis methods [A, B, C, D], or human evaluation.

[A] Modern Text Hiding, Text Steganalysis, and Applications: A Comparative Analysis; Entropy. 2019
[B] Linguistic Steganalysis With Graph Neural Networks; IEEE Signal Processing Letters. 2021
[C] A fast and efficient text steganalysis method; IEEE Signal Processing Letters. 2019
[D] Exploiting language model for efficient linguistic steganalysis; ICASSP 2022


**Summary Of The Paper:**

This work proposes to view the task of steganography as a minimum entropy coupling problem. The authors show that (1) coupling algorithms yield steganography procedures with perfect security; (2) the approach induced by minimum entropy coupling maximizes information throughput; (3) apply the gained insights to steganography with deep generative model covertext distributions.

**Summary Of The Review:**

I believe that this work holds some novelty for a subgroup in the ML community. While the assumptions made in this work heavily rely on the assumptions made by Cachin might not hold in practical scenarios, the authors provide a “secure” steganography approach from a theoretical standpoint. The authors also showed that their approach is effective in some practical scenarios.

---

> ### Author Response · Authors · 2022-11-11
> **Author Response**
>
> Thanks for your review! We agree with the reviewer's characterization of our submission, other than the third weakness. We provide comments regarding the weaknesses below.
>
> - This work relies on the assumption that “the exact distribution of covertext is assumed to be known”. However, in practice, this might not be the case, since the model of language might not be perfectly known. Hence, for most practical scenarios this approach might not provide perfectly secure steganography.
>
> We agree that this is a strong assumption and that, in practice, this might not be the case. However, we would point out here that there is a large body of literature on Cachin's model that makes the same assumption. Thus, we feel that there is sufficient interest in the setting among the community, even if it might not capture some practical scenarios.
>
> - While the results in Figure 2 are close to zero, they are not exactly zero. Does this mean that perfect security is not obtained as claimed?
>
> The non-zero results are a byproduct of finite numerical precision and could be made arbitrarily small by using higher levels of numerical precision. We would characterize the situation as analogous to how gradient descent provably converges on strongly convex functions but, on actual hardware, can only converge to a neighborhood dictated by the level of numerical precision.
>
> - The authors evaluated the security of their approach empirically, however, it is not clear if this method can withstand modern steganalysis methods [A, B, C, D], or human evaluation.
>
> We feel that, for methods with perfect security guarantees, the outcome of such experiments is already clear.
>
> If the true covertext distribution is accessible, perfect security guarantees that the stegotext and covertext are statistically indistinguishable; thus, in such a setting, iMEC would be undetectable by both steganalysis methods and human evaluation. If the reviewer is aware of open source code for steganalysis, we are happy to add experiments demonstrating this claim. (We do not see any link to code in [A, B, C, D].)
>
> On the other hand, if only an approximate, rather than exact, covertext distribution is available, then using a method with perfect security reduces the steganalysis problem to distinguishing between the approximate covertext distribution and the exact covertext distribution. To us, this direction feels orthogonal to the contribution of the submission.

---

### Official Review · Reviewer_yanH · 2022-10-22

**Confidence:** 3
**Correctness:** 4
**Technical Novelty And Significance:** 4
**Empirical Novelty And Significance:** 3
**Recommendation:** 8

**Clarity, Quality, Novelty And Reproducibility:**

The paper is very clear with a high overall quality.
The results are quite novel and reproducibility does not seem to be an issues.


**Details Of Ethics Concerns:**

There are no ethical issues to address.

**Strength And Weaknesses:**


Strengths:
1) The paper provides thorough theoretical derivations for this increasingly interesting and important topic
2) The paper is very clear and methodologically correct
3) The fact that a theoretical constraint is also efficient is a very interesting result of this paper

Weaknesses:
1) The fact that the paper assumed the Cachin (1998) model with transmission being made by a noise-free (perfect) channel is, as pointed out by the authors, a drawback that limits the importance and applicability of these results.
2) Nowadays, important applications of steganography are not limited to 1D signals and thus application on 2D images are of extreme importance, opening relevant applications. I suggest the authors briefly discuss this in the paper, particularly the applicability of the paper results to images.


**Summary Of The Paper:**

This paper presents theoretical proofs of perfectly secure (zero KL divergence) steganography, particularly that it depends on being induced by a coupling (necessary and sufficient condition) and that for efficiency it will also depend on a minimum entropy coupling (that benefit from very recent advances in the approximation of minimum entropy coupling iteratively). The paper presents experiments showing practicality of the assumptions in a (unrealistic) noise-free channel.

**Summary Of The Review:**

The paper is a theoretical paper with thorough derivations and very clear discussions. The result impact is somehow limited to the assumptions of a noise-free channel, but still important.

---

> ### Author Response · Authors · 2022-11-11
> **Author Response**
>
> Thanks for your review! We agree with the reviewer's characterization of our submission on both the strengths and the weaknesses. Regarding the weaknesses:
>
> > The fact that the paper assumed the Cachin (1998) model with transmission being made by a noise-free (perfect) channel is, as pointed out by the authors, a drawback that limits the importance and applicability of these results.
>
> We agree that assuming a noise-free channel is a strong assumption. We have a few additional comments regarding this assumption:
> - First, this assumption is made by the original framework that Cachin introduced for information theoretic steganography and by existing work for steganography with deep generative models (including all of our baselines).
> - Second, because the covertext channel is generally chosen to be an innocuous medium of communication, it may actually be realistic to assume that the channel is noise free or close to noise free in some situations since communicators prefer communication mediums that are as close to noise free as possible.
> - Third, it is straightforward to combine existing approaches to noise free steganography (i.e., both iMEC and the baselines we considered) with *error detection* mechanisms. (Note error detection is weaker than error correction.) Thus, it would at least be safe to deploy these approaches in settings in which there is a risk of channel noise.
>
> > Nowadays, important applications of steganography are not limited to 1D signals and thus application on 2D images are of extreme importance, opening relevant applications. I suggest the authors briefly discuss this in the paper, particularly the applicability of the paper results to images.
>
> We agree that 2D images are an important application. However, we would like to point out that both iMEC and other existing methods can be directly applied to 2D image mediums by using autoregressive image models, such as (Generating Long Sequences with Sparse Transformers; Child 2019). To illustrate this point, we are working on such experiments now. If we are able to complete them prior to the end of the discussion period, we will update the submission. Otherwise, if the submission is accepted, **we commit to having such experiments included in the camera ready.**

---

### Official Review · Reviewer_tJp6 · 2022-10-25

**Confidence:** 5
**Correctness:** 2
**Technical Novelty And Significance:** 1
**Empirical Novelty And Significance:** 1
**Recommendation:** 1

**Clarity, Quality, Novelty And Reproducibility:**

Lacks clarity and novelty is questionable. Please note the weaknesses section in the review.

**Details Of Ethics Concerns:**

No ethical concerns as of yet.

**Strength And Weaknesses:**

Strength::
Security is an important topic in computing and should be pursued.

Weakness::
The paper can be improved based on the following points.
1) The title of the paper is not appropriate as it is misleading and should be changed. In security field, there is no such thing as "perfectly secure" mechanism. Every security measures has its own Pros and Cons depending on circumstances. The title should be revised.
2) In section 1, it is mentioned, "A procedure is said to be perfectly secure if it guarantees a divergence of zero." This is only true in theoretical sense and not in practical. So, this should be made clear in the statement with appropriate citations.
3) The motivation in the section 1 is not strong enough. Provide a motivational case study to engage the readers and to prove the importance of such research.
4) Section 2 (Background) is a bit misleading in written format as it mostly consist of problem formulation rather than background research. Please rename the title and provide background research on other existing related methods.
5) In section 3 & 4, please define each variables used/mentioned for readability proposes. Not everyone who would be reading this paper would be a field expert in computer security.
6) Section 5 (experimental evaluation) is weak. Comparative study is lacking and should be included to show the efficacy of the claims being made. Pleas provide that.
7) In a conference paper, it might be better to segregate future works into a separate section along with discussions. And also have a separate conclusion section.
8) The paper seems a bit out of scope for this conference and might be better tuned towards computer security related conferences.

**Summary Of The Paper:**

In this paper, the authors show that a steganography procedure is perfectly secure under Cachin (1998)’s information theoretic-model of steganography if and only if it is induced by a coupling. Moreover, the authors show that, among perfectly secure procedures, a procedure is maximally efficient if and only if it is induced by a minimum entropy coupling.

**Summary Of The Review:**

1) The title of the paper is not appropriate as it is misleading and should be changed. In security field, there is no such thing as "perfectly secure" mechanism. Every security measures has its own Pros and Cons depending on circumstances. The title should be revised.
2) In section 1, it is mentioned, "A procedure is said to be perfectly secure if it guarantees a divergence of zero." This is only true in theoretical sense and not in practical. So, this should be made clear in the statement with appropriate citations.
3) The motivation in the section 1 is not strong enough. Provide a motivational case study to engage the readers and to prove the importance of such research.
4) Section 2 (Background) is a bit misleading in written format as it mostly consist of problem formulation rather than background research. Please rename the title and provide background research on other existing related methods.
5) In section 3 & 4, please define each variables used/mentioned for readability proposes. Not everyone who would be reading this paper would be a field expert in computer security.
6) Section 5 (experimental evaluation) is weak. Comparative study is lacking and should be included to show the efficacy of the claims being made. Pleas provide that.
7) In a conference paper, it might be better to segregate future works into a separate section along with discussions. And also have a separate conclusion section.
8) The paper seems a bit out of scope for this conference and might be better tuned towards computer security related conferences.

---

> ### Author Response · Authors · 2022-11-11
> **Author Response**
>
> 1. In security field, there is no such thing as "perfectly secure" mechanism.
>
> As is made clear in the submission, we use the term "perfectly secure" as was defined by Cachin (1998) and as it has been used in extensive follow up literature.
>
> 2. A procedure is said to be perfectly secure if it guarantees a divergence of zero." This is only true in theoretical sense and not in practical.
>
> This is the definition of perfect security within Cachin (1998)'s model. It is a definition -- not a statement with a true or false value. To the extent that the reviewer is pointing out that Cachin's model makes strong assumptions, we agree and note that this was discussed in the Conclusions and Future Works section.
>
> 4. Section 2 (Background) is a bit misleading in written format as it mostly consist of problem formulation rather than background research.
>
> We introduce the problem setting in the background section and discuss the related work in the related work section.
>
> 5. In section 3 & 4, please define each variables used/mentioned for readability proposes. Not everyone who would be reading this paper would be a field expert in computer security.
>
> We believe that we did define each variable -- note that some definitions are in the background (section 2). If we are mistaken, please let us know specifically which variable's definition we omitted.
>
> 6. Section 5 (experimental evaluation) is weak. Comparative study is lacking and should be included to show the efficacy of the claims being made.
>
> Section 5 includes a comparison against the relevant baselines of which we are aware. We feel that it shows the efficacy of our approach. If the reviewer is aware of other relevant baselines, please let us know.
>
> 7. In a conference paper, it might be better to segregate future works into a separate section along with discussions. And also have a separate conclusion section.
>
> The first paragraph in Section 7 discusses conclusions; the second paragraph in Section 7 discusses limitations and future work.
>
> 8. The paper seems a bit out of scope for this conference and might be better tuned towards computer security related conferences.
>
> The existing literature for steganography with deep generative models is mostly in NLP conferences (such as ACL and EMNLP), rather than security conferences, so we feel that it is not out of scope.

---

### Official Review · Reviewer_x2x4 · 2022-10-25

**Confidence:** 2
**Correctness:** 3
**Technical Novelty And Significance:** 3
**Empirical Novelty And Significance:** 3
**Recommendation:** 6

**Clarity, Quality, Novelty And Reproducibility:**

- The paper is well-written.
- The proposed method seems novel, i.e., the first instance of steganography algorithm with non-trivial efficiency and perfect security guarantees.
- Code is provided for reproducibility.

**Details Of Ethics Concerns:**

N/A.

**Strength And Weaknesses:**

The paper is well-written. The authors start by formulizing and explaining the relationship between perfect security and couplings of distributions through two results. Then, the authors accordingly introduce the first instance of steganography algorithm with non-trivial efficiency and perfect security guarantees.

Since steganography is not my primary research area, the theoretical part of this paper is beyond my knowledge. So, I cannot comment its theoretical contribution reaches the standard of ICLR. But it seems to me that the proposed method is well-motivated, based on two findings that the perfect steganographic security is, in fact, equivalent to a coupling problem and achieving maximal transmission efficiency among perfect security procedures is equivalent to a minimum entropy coupling problem. So, I rate this paper as borderline accept for now and will refer to the comments of other reviewers for my final rating.


**Summary Of The Paper:**

In this paper, the authors formalize the relationship between perfect security and couplings of distributions, and propose the first instance of a steganography method that has the benefits of non-trivial efficiency and perfect security guarantees. Experiments with GPT-2 and WaveRNN demonstrate the effectiveness of the proposed method. Also, the code is provided for reproducibility.

**Summary Of The Review:**

Steganography is not my primary research area. It seems to me that the paper has a solid theoretical support.

---

> ### Author Response · Authors · 2022-11-11
> **Author Response**
>
> Thanks for the review! We agree with the reviewer’s characterizations of our work. We are happy to answer any questions that may come up.

---

### Decision · Program_Chairs · 2023-01-20

**Decision:**

Accept: poster

**Justification For Why Not Higher Score:**

The paper makes interesting contributions but does have some flaws so does not merit an oral presentation based on the scores and my reading of the paper

**Justification For Why Not Lower Score:**

I am choosing to disregard one reviewer which clearly seemed to have made quite a few mistakes this puts the average score at 6.7 making it a non-borderline accept paper.

**Metareview: Summary, Strengths And Weaknesses:**

In this work, the authors show that a steganography procedure has "perfect" security under Cachin's information theoretic-model of steganography if and only if it is induced by a coupling. They also show that among perfectly secure procedures, a procedure is maximally efficient if and only if it is induced by a minimum entropy coupling. The authors also develop steganography algorithms that achieve perfect security guarantees with non-trivial efficiency and are highly scalable. Empirical validation is provided by comparing a minimum entropy coupling-based approach to three modern baselines using GPT-2 and WaveRNN as communication channels. The results suggest that it may be natural to view information-theoretic steganography through the lens of minimum entropy coupling. Most reviewers thought the paper is well-written, the method is novel, and liked the fact that this is the first instance of steganography algorithm with non-trivial efficiency and perfect security guarantees. The reviewers raised concerns about using the Cachin model, restriction to 1D, perfectly secure definition and other issues. In my opinion the authors provided an adequate response to the reviewers. I recommend acceptance.







**Note From Pc:**

if the above contains the word "oral" or "spotlight" please see: "oral" presentation means -> notable-top-5% and "spotlight" means -> notable-top-25%. As stated in our emails, we are disassociating presentation type from AC recommendations

**Summary Of Ac-Reviewer Meeting:**

This is technically borderline but I am choosing to dis-regard one reviewer which then puts this paper clearly into the accept column. Hence, I did not think a meeting/discussion was necessary.